# Epithelial-Mesenchymal Transition: A Fundamental Cellular and Microenvironmental Process in Benign and Malignant Prostate Pathologies

**DOI:** 10.3390/biomedicines12020418

**Published:** 2024-02-11

**Authors:** Aviv Philip Goncharov, Nino Vashakidze, Gvantsa Kharaishvili

**Affiliations:** 1Department of Clinical and Molecular Pathology, Palacky University, University Hospital, 779 00 Olomouc, Czech Republic; avivphilip.goncharov01@upol.cz (A.P.G.); nino.vashakidze01@upol.cz (N.V.); 2Department of Human Morphology and Pathology, Medical Faculty, David Tvildiani Medical University, Tbilisi 0159, Georgia

**Keywords:** epithelial-mesenchymal transition, EMT, prostate cancer, BPH, transcription factors

## Abstract

Epithelial-mesenchymal transition (EMT) is a crucial and fundamental mechanism in many cellular processes, beginning with embryogenesis via tissue remodulation and wound healing, and plays a vital role in tumorigenesis and metastasis formation. EMT is a complex process that involves many transcription factors and genes that enable the tumor cell to leave the primary location, invade the basement membrane, and send metastasis to other tissues. Moreover, it may help the tumor avoid the immune system and establish radioresistance and chemoresistance. It may also change the normal microenvironment, thus promoting other key factors for tumor survival, such as hypoxia-induced factor-1 (HIF-1) and promoting neoangiogenesis. In this review, we will focus mainly on the role of EMT in benign prostate disease and especially in the process of establishment of malignant prostate tumors, their invasiveness, and aggressive behavior. We will discuss relevant study methods for EMT evaluation and possible clinical implications. We will also introduce clinical trials conducted according to CONSORT 2010 that try to harness EMT properties in the form of circulating tumor cells to predict aggressive patterns of prostate cancer. This review will provide the most up-to-date information to establish a keen understanding of the cellular and microenvironmental processes for developing novel treatment lines by modifying or blocking the pathways.

## 1. Description of Types of EMT

Epithelial-mesenchymal transition (EMT) is a complex manifestation of epithelial plasticity [1]. EMT is a process that allows epithelial cells, which are typically attached to the basement membrane, to pass through various changes that enable them to acquire a mesenchymal cell phenotype. The process includes changes in cytoskeleton and cell shape, enhancing migration and establishment of metastatic potential, invasiveness, increased resistance to apoptosis, and significantly elevated production of extracellular matrix (ECM) components [2].

EMT was recognized in the late 70s and received distinct attention due to its physiological presence, such as embryonic development and pathological conditions. Over the years, a significant focus has been on the relationship between EMT and various cancers. 

The current understanding of the signaling processes that mediate EMT is starting to provide new opportunities for utilizing the underlying molecular mechanisms to create novel treatments.

On the other hand, the reverse process, mesenchymal to epithelial transition (MET), also occurs and is essential in illustrating the potential of EMT to be a reversible process. Also, endothelial cells show properties similar to epithelial cells and can lose endothelial characteristics while gaining mesenchymal characteristics. 

EMTs are related to three physiological and pathological scenarios with different consequences.

### 1.1. Type 1 EMT

Type 1 EMT is associated with embryogenesis and occurs at a few locations and stages in the process of organ development, e.g., during the gastrulation when ectodermal cells give rise to mesoderm and also during neural crest migration. Type 1 EMT generates new tissue with mesenchymal phenotype but was not found to cause other manifestations such as fibrosis or systemic invasion by high-grade tumors [3,4].

The coordination of EMT is enabled by various proteins such as Snail, Eomesodermin, and Mesoderm posterior protein. The Snail suppresses the E-cadherin, thus promoting the transition. This explains the origin of migratory neural crest cells from the neuroectodermal epithelial cells [5]. 

Developmental EMT is regulated by morphogenic signaling pathways [4]. A common path to the EMT and gastrulation is the Wnt signaling pathway. Wnt plays an essential role in morphogenesis, cellular orientation, and organization. The pathway is activated at the posterior region of the embryo, resulting in the creation of the primitive streak. This is the same pathway that assists with the initiation of EMT. Later, after EMT facilitates the formation of the primitive streak, the opposite process, MET, gives rise to other components, such as the peripheral nervous system and adrenal medulla cells. Another interesting example is the formation of nephron epithelium in the kidney. Mesenchymal cells aggregate around the ureteral bud and acquire cell-to-cell adhesion properties by MET [6].

### 1.2. Type 2 EMT

The second type of EMT is associated with wound healing, tissue regeneration, and organ fibrosis. It begins as part of a regular healing event that typically creates fibroblasts and other cells to recreate tissues after trauma or an inflammatory process [3]. In contrast to type 1 EMT, these types are associated with inflammation and cease when it is attenuated. In chronic inflammation, the abnormal formation of myofibroblasts is associated with permanent progressive fibrosis, leading to organ failure due to high deposition of extracellular matrix components such as collagens, laminins, elastin, and tenascins [5]. Yin et al. showed that nintedanib (Ofev), a medication used nowadays mainly for the treatment of pulmonary fibrosis, may also be used for proliferative vitreoretinopathy (PVR) by preventing TGF-β2-induced EMT in retinal pigment epithelial cells and thus providing a new therapeutical target for PVR [7]. 

As will be shown later in this review, few signaling pathways promote EMT in tumors, but nevertheless, it has been shown that these molecules, such as miRNA, ZEB, and TGF-β2, play a crucial role in abnormal scarring. Moreover, several studies showed the difference in miRNA expression between normal skin and hyperplastic scars, showing higher expression in the scarred tissue and a possible correlation with TGF-β2 signaling [5]. 

Until now, the number of medications and therapeutic possibilities for fibrosis is insufficient. For instance, there are only two FDA-approved drugs for pulmonary fibrosis (nintedanib and pirfenidone) [8]. In addition to natural substances that demonstrated the ability to alter the EMT, such as honey, curcumin, *olea europea*, and *Paeonia lactiflora*, tissue engineering is currently focusing on creating new treatment lines based on understanding the EMT process. One of the treatment possibilities suggested is polyacrylamide hydrogels, which are matrices with different properties and are known to modulate EMT. Such biomaterials possess different biochemical and biophysical properties, thus facilitating the physiological process of tissue regeneration [5].

### 1.3. Type 3 EMT

Type 3 EMT occurs in neoplastic cells that pass through genetic changes affecting oncogenes and tumor suppressor genes. Carcinoma cells undergoing this type of EMT may invade and metastasize and thus favor cancer progression. Features of EMT have been observed in various kinds of tumors, for example, in breast [4], prostate [9], ovarian [10], colon [11], and other cancers. Gavert et al. suggest a hypothesis, and according to it, developmental programs are reactivated during tumorigenesis, contributing to the overall cellular outcome. Many regulation checkpoints of the EMT process are highly expressed and are associated with a non-coordinated and less ordered process compared to the developmental EMT process [12]. 

Interestingly, an association was found between low expression of specific molecules such as E-cadherins and membranous β-catenin and the progression of prostate cancer. Furthermore, it has been shown that the EMT process, also known as cellular plasticity, is the cause of metastatic disease mediated by the cells that act as cancer stem cells [13]. Rusetti et al. showed on a mouse model that the cells that possess cellular plasticity are more prone to survive in distant regions other than the primary tumor, while those that did not pass through EMT could initiate a primary tumor but would not pass through further metastasis [14]. Jiaren Li et al. showed in different research that the accumulation of mesenchymal cells is strongly associated with benign prostatic hyperplasia (BPH). Therefore, he suggests using EMT as a potential target for treating BPH [15]. 

Other research showed, by protein expression analysis, that EMT in bone metastasis from prostate cancer creates cells that are often resistant to chemotherapeutic treatment due to their arrest in the G0 state while contacting bone marrow stromal cells. It has been shown that TGF-β, a potent activator of EMT and metastasis, plays a role in the induction of chemoresistance, as described [16]. In a work by Lu et al. on a 3D model system, it was found that TGF-β2 was highly induced in a co-culture with THP-1 cells. Hence, they tried to use an anti-TGF-β2 neutralizing antibody at various concentrations to determine the concentration that could inhibit macrophage-mediated growth of BPH-1. They found that anti-TGF-β2 neutralizing antibodies at a concentration of 50 ng/mL could inhibit the growth of BPH cells. Similarly, the anti-TGF-β2 neutralizing antibody attenuated the EMT induction marker (N-cadherin) expression. It increased the expression of E-cadherin, suggesting that TGF-β2 has a role in BPH formation, and neutralization of TGF-β2 may block the EMT that occurs in the pathogenesis of BPH [17]. 

## 2. Cellular and Tissue Morphology Characterizing EMT

Polarized epithelial cells, which are usually organized in stratified or single layers and acquire mesenchymal properties, have the capability of locomotion [18]. The fundamental events that impact the EMT process include a reduction in cell-cell adherence by repression and a change of location of cadherins, occludin, claudin, and desmoplakin [19] (Figure 1). 

β-catenin is an E-cadherin-supporting molecule and frequently translocates from the cellular membrane to the nucleus to initiate further EMT events [20,21]. Circumferential F-actin fibers of the cytoskeleton are replaced by a network of stress fibers at the tips of which ECM adhesion molecules (including integrins, paxillin, and focal adhesion kinase) localize [22]. These changes allow cells to separate, leave the basement membrane, and lose the cellular apicobasal polarity, which is typical in epithelial cells, acquire a front-back polarity [23], and gain a different morphology, which is more variable, fibroblast-like cell shape for the facilitation of cell movement. Fibroblastoid cells play a role in the degradation or synthesis of ECM via different molecular pathways [18].

Tumor cells presenting the properties and the phenotype of mesenchymal cells show more invasiveness, metastatic capabilities, resistance to chemo and endocrine therapy, resistance to radiation-induced DNA damage, increased interaction with stromal inflammatory mechanisms, and increased cell survival [24,25].

A crucial part of the process is the change in the expression of various types of ECM components. For example, the expression of epithelial intermediate filaments such as β4 integrin and ZO-1 is reduced, while the equivalent mesenchymal proteins, such as vimentin, N-cadherin, and α-SMA, are typically increased. Matrix metalloproteases of different types (MMP-1, -2, -3, -7, and -14) are upregulated, thus allowing the cells to detach from each other and invade the basement membrane. The cells must undergo phenotypic changes to pass through the molecular modifications [26]. 

## 3. EMT Induction and Mechanism

EMT is induced by cytokines capable of proteolytic digestion of basement membranes on which the epithelial cell is attached. The process can be initiated by several oncogenes, including *RASV12* [27], *ErbB2* [28], and *TRKB* [29,30]. They activate many other components, such as PI3, MAP kinases, Notch, Wnt, and NF-κB pathways, all involved in EMT regulation [31]. Many other transcription factors, such as Snail 1, Snail 2, Twist, δEF1 (ZEB 1), SIP-1 (ZEB 2), and E12/E47, have been shown to induce the process of EMT. However, whether these factors function independently or combined with other elements to activate the EMT process is unknown. 

Hierarchy exists in the expression of the factors. Snai1 is expressed in the early phases of the process, while other factors such as Snai2, Zeb1, and Twist are introduced later as part of the regulation and maintenance of migratory capabilities. Their role in EMT will be discussed later. 

Growth factors such as TGF-β, EGF, IGF-II, FGF-2, and HGF are locally expressed and facilitate the process by binding epithelial receptors with specific intrinsic kinase activity. HGF, the ligand of the c-met receptor, was proclaimed in 1985 as the first EMT inducer molecule [32]. As a result, the tumor also starts to produce proteolytic enzymes. It remodels ECM and the basement membrane and prepares the environment to be suitable for migration and invasion.

Grant et al. describe that a hormonal-mediated axis regulates prostate cancer and that the early stages of the tumor’s proliferation are dependent on androgens. The tumor alters several signaling pathways for achieving invasive and aggressive capabilities, such as the Androgen receptor (AR) pathway and the EMT process. Therefore, many clinical trials and FDA-approved medications target the tumor via AR pathways, such as abiraterone acetate, a type of CYP17A1 inhibitor that blocks the synthesis of androgens, thus affecting tumor growth. The researchers emphasize that during embryogenesis, the prostate gland is formed and developmentally regulated by SRY-related high-mobility-group box (Sox) transcription factors. SOX9 is one of the factors which is shown to be significantly elevated in recurrent prostate cancer. Apparently, SOX9 takes part in Wnt/β-catenin and FGF signaling pathways that induce AR expression and EMT. The cells acquire the properties crucial for further metastasis [33]. Another interesting induction mechanism of EMT in prostate cancer is the heat shock protein (HSP), which will be discussed in Section 5.

An important induction mechanism was described by Lemster et al., according to which KDM5C modulates EMT signaling pathways such as Hedgehog, Wnt, Notch, PI3K-Akt-mTOR and other factors such as ZEB1, ZEB2, SNAI2, and others. KDM5C is a molecule that demethylates H3K4 histones. It has been shown that knockdown of KDM5C is associated with reduced risk for metastasis by high expression of E-cadherin. Also, KDM5C regulates the TGF-β signaling cascade by changes in SMAD7 expression. Therefore, inhibition of KDMC5 showed significant downregulation of SMAD7 and decreased TGF-β expression and EMT [34].

The TGF-β signaling pathway is also associated with converting fibroblasts into cancer-associated fibroblasts (CAF), which act as promoters of EMT. According to Wu et al., suppression of TGF-β by silibinin inhibits the differentiation of fibroblasts into CAF, suppresses the expression of vimentin and α-SMA, and thus, decreases the invasiveness of prostate cancer [35].

## 4. EMT Proteome and Genome; Molecular Switch

EMT proteome reflects the change in components, whether gained or lost, during the conversion of epithelial cells into mesenchymal cells in physiological cases or the transition of tumor epithelium to metastatic cells.

Studies on the EMT process are based mainly on the information acquired from proteins located in the epithelia rather than in fibroblasts or tumor cells. Many proteins are gained or maintained in the process and are described in Figure 1.

### 4.1. Twist

Twist is a transcription factor that plays an integral role in embryogenesis and is a crucial EMT regulator in cancers [36]. Twist and Snail1 upregulate Zeb1, leading to the downregulation of E-cadherin [37]. The ectopic expression of Twist in canine kidney cells caused a decrease in E-cadherin, α-, β- and γ-catenins, and therefore, acquirement of mesenchymal markers vimentin, fibronectin, smooth muscle actin, and N-cadherin [36]. 

Twits have been found to induce EMT in Hela and MCF7 cells, accompanied by the upregulation of ALDH1 and CD44, which are known as markers of stem cells [38]. In gastric cancer, the tumor stage and depth of invasion significantly correlate with the expression of Twist mRNA [39]. In vivo studies suggested that elevated Twist expression might be responsible for breast cancer lung metastases [36]. Moreover, it has been found that high expression of Twist is associated with high pre-treatment prostate-specific antigen (PSA) levels, high Gleason score (>7), advanced tumor stage, involvement of lymph nodes, distant metastasis, and biochemical progression. Also, a significant association was found between high Twist-1 and shorter biochemical progression-free survival [40]. 

In another article, it was described that the increased expression of Twist was in a primary tumor, while expression was reduced in distant metastasis. Also, a strong Twist expression was detected along with Slug and was associated with HIF-1α in localized prostate cancer, and a strong expression of twist was associated with HIF-1α in castration-resistant prostate cancer [41]. It was shown by Jin et al. that a protein known as trophinin-associated protein (TROAP) is overexpressed in prostate cancer and promotes its progression. At the same time, low levels can inhibit cancer cell proliferation and cause arrest in the cell cycle. Interestingly, they showed that TROAP knockdown induced apoptosis and prevented migration and invasion abilities by inhibiting the Twist/c-Myc pathway. On the other hand, overexpression of Twist showed a partial decrease in the inhibition of cellular proliferation induced by the silencing of TROAP [42]. 

### 4.2. Snail Family

Snails are essential for the induction of EMT. Snail1 is vital in forming the mesoderm layer and neural crest during embryogenesis and is an important factor in EMT-associated tumor progression. For example, Snail induces EMT in pancreatic cancer cells by expressing vimentin and suppressing E-cadherin. Moreover, Snail was found to be closely related to tumor growth properties, invasion, and metastasis [43]. 

In breast cancer, Snail1 is associated with tumor dedifferentiation. In invasive ductal carcinoma (IDC), Snail was observed in 47% of the cells and was expressed in most grade 3 tumors and more than half of grade 2 tumors but not in any grade 1 IDC [44]. The knockdown of Snail2 caused a full blockage of Twist1’s property to inhibit the transcription of E-cadherin [45]. Twist1 binds to a Snail 2 promoter to induce its transcription. The latter’s induction is essential for Twist 1-induced cell invasion and distant metastasis in mice. The expression of Twist 1 and Snail 2 is highly correlated in breast, bladder, and esophageal squamous cell carcinomas [44,46,47]. Gene expression analysis of CD44+/CD24− breast cells compared to CD44−/CD24+ cells revealed increased expression of 32 EMT-associated genes, including SLUG, ZEB-1, ZEB-2, periostin, Hedgehog signaling associated gene Gli-2, and the metastasis-associated gene SATB-1 [48]. Transgenic overexpression studies showed that only SLUG could change the phenotype of CD44−/CD24+ MCF-10A cells in favor of the induction of a subpopulation of CD44+/CD24− cells. In the luminal type of breast cancer cell line MCF-7, the overexpression of SLUG created cells with a CD44+/CD24+ phenotype, suggesting that basal cell types and not luminal types are more susceptible to acquisition of CD44+/CD24− phenotype, which is associated with EMT. Also, other specific EMT-associated genes inducing a CD44+/CD24− phenotype in MCF-10A cells are N-cadherin, FHL-1, ST-2, Wnt5B, FOXC2, ETV5, SATB-1, SLUG, and Gli-2. For example, it has been shown that the upregulation of the NF-κB subunit of p65 also upregulated ZEB-1 and ZEB-2 gene expression [49], which resulted in an increase in CD44+/CD24+ cells but not in CD44+/CD24- MCF-10A cells [48].

Katoh and Katoh have identified another member of the Snail family, Snail3. Human SNAI3 (Snail3) mRNA was expressed in skin melanoma, lung squamous cell carcinoma, and germ cell tumors. Authors have suggested that Snail/Gfi-1 (SNAG) zinc-finger proteins act as transcriptional repressors and are important in tumorigenesis and embryogenesis. Thus, it is suggested that these proteins, especially SNAI3, may be used as a promising and potent pharmacological target [50]. There is not enough experimental data on the Snail3 function [51].

Regarding prostate cancer, Snail has been associated with resistance to cisplatin and tumor necrosis factor-related apoptosis-inducing ligand (TRAIL), as there is a process of sensitization of the cells to cisplatin and TRAIL-mediated apoptosis. Moreover, the effect of Snail is an increase in migration and invasion. It has been shown on PTEN knockout mice with developed prostate cancer that there is a correlation between the expression of Snail and the stage of the tumor. Thus, there is low expression in benign prostatic metaplasia and high expression in bone metastatic specimens [51]. 

Snail overexpression correlates with the progression and disease stage of prostate cancer in vivo as well. Its activity is regulated by different growth factors such as TGF-β1, EGF, and vascular endothelial growth factor-A (VEGF-A). VEGF-A and TGF-β promote Snail nuclear localization in prostate cancer cells, while TGF-β and EGF induce EMT in prostate cancer cells, leading to their distinct metastatic potential. The mechanism by which VEGF-A contributes to the initiation of EMT is by binding to its receptor neuropilin-1 (NRP1), which then promotes nuclear localization of Snail. Surprisingly, there have been other factors that may express the overactivity of Snail, e.g., a hypoxic state. Hypoxia can influence EMT by regulation of VEGF-A expression and stimulation of the VEGF-A/NRP1 pathway that results in Snail nuclear localization and EMT in prostate cancer cells [51].

### 4.3. Zeb1 and Zeb2

The Zeb family consists of two essential members: Zeb1 (also known as TCF8 and δEF1) and Zeb2 (ZFXH1B and SIP1) [52]. The members of this family interact with the DNA by simultaneously binding the two zinc-finger domains to E-boxes [52]. Both proteins are potent repressors of CDH1 (E-cadherin gene). Although they are not as potent as Snail in the induction of EMT or in the repression of CDH1 in the in vitro assays [53], their silencing, especially that of Zeb1, has a higher impact on CDH1 expression than Snail [52,54]. Inhibition of Zeb1 and 2 by the miR200 family restores E-cadherin protein expression [55].

Snail1 and Slug activate Zeb family members, TCF3, TCF4, Twist, Goosecoid, and FOXC2, which may all be important in maintaining the EMT phenotype [56]. Docetaxel, a mitotic inhibitor chemotherapeutic, remains an essential line of treatment for advanced castration-resistant prostate cancer, but in some cases, resistance to the chemotherapeutic occurs. Hanrahan et al. showed an interesting connection between overexpression of EMT-related transcription factors from the ZEB family and the resistance of prostate cancer to Docetaxel. The repression of E-cadherins mediated the resistance and the EMT. When a knockdown of ZEB1 and 2 was performed, the E-cadherin expression increased markedly [57]; therefore, this pathway may be a potential line of treatment in docetaxel-resistant tumors.

### 4.4. MicroRNAs

Noncoding microRNAs are important signaling factors in EMT regulation [58]. MicroRNA 200 (miR200) and miR205 cause the inhibition of ZEB1 and ZEB2, which act as the repressors of E-cadherin and, thus, play a role in maintaining the epithelial cell phenotype [59]. In breast carcinoma, a loss of miR200 induces EMT via decreased E-cadherin and increased vimentin expression [59]. On the contrary, miR21 is upregulated in many other types of cancers and facilitates TGF-β–induced EMT [2]. It has been shown that miR200-mediated down-regulation of WAVE3, a metastatic favorable protein, significantly reduces the metastatic abilities of breast and prostate cancer cells [60]. A change in the cellular phenotype from mesenchymal to epithelial was achieved by inhibiting WAVE3 expression or by overexpression of miR200b or siRNA and indicates the role of WAVE3 in this process. Nevertheless, it is still unknown whether WAVE3 is directly involved in the ZEB1/ZEB2/E-cadherin pathway [60]. Sharma et al. explained that the molecular basis behind the overexpression of microRNA is the alteration in the copy number of microRNA, epigenetic modifications, upregulated expression of Dicer, a mutation in stem regions of pre-miRNA, single nucleotide polymorphism, and androgen receptor-regulated mechanisms [61].

Fabris et al. showed in a review that the microRNAs involved in almost any pathway and cellular process may be used as a potential marker in clinical practice. The fact that it can be found even in urine and semen samples that are highly correlated with the existence of prostate cancer opens a window to new diagnostic methods, especially since it has been suggested that both PSA and DRE (digital rectal examination) are neither specific nor sensitive enough. Fabris’s review showed a linear correlation between microRNA levels and cancer risk. The subtype of microRNA that was found to be constantly altered in the serum of prostate cancer patients is miR181 and is associated with EMT and apoptotic pathways. Interestingly, the review described miR96 as a marker that correlates with the survival of patients after radical prostatectomy and as a marker of biochemical recurrence, demonstrating that tumors expressing a high amount of miR96 have a significant decrease in recurrence-free survival [62]. Sekhon et al. showed that the loss of EMT-inhibiting microRNA or a gain of EMT-promoting microRNA leads to the induction of prostate cancer and to further progression, metastasis, and recurrence [63].

## 5. Heat Shock Proteins and Prostate Cancer

HSPs are a group of proteins from the family of chaperones. Morphologically, the HSPs resemble a barrel into which proteins that pass through denaturation are redirected for the reversible process. There are a few types of HSPs-HSP100, 90, 70, 60, 40, and small HSP/α-crystallins. The proteins are located in different cell loci and are associated with cytoprotection, providing cellular tolerance to damage, thermoresistance, etc. [64,65]. The need for the tumor in HSPs is understandable. The tumor faces a hostile microenvironment that is acidic, hypoxic, and nutrient-deprived. In such an environment, damage to cellular compartments is inevitable, and the role of HSPs is crucial in maintaining the cell’s relative homeostasis [66]. Nevertheless, HSPs play a role that is much more complicated than only maintaining the normal intrinsic environment. 

The association between the expression of HSPs and prostate cancer was described by Fu et al. as factors that promote the EMT. Generally, HSP70 and HSP90 are known to be involved in prostate cancer and may be divided into subgroups according to their localization in the cytosol, mitochondria, and endoplasmic reticulum. As discussed earlier, an essential part of tumorigenesis in prostate cancer is the androgen receptor (AR). Thus, the course of the disease may be altered by using drugs that will modify the AR, such as finasteride. It has been found that HSP70 supports tumorigenesis by activating AR. Consequently, the inhibition of HSP70 will result in the inhibition of AR, and prostate tumor growth will be halted, thus improving the therapy in castration-resistant prostate cancer (CRPC) patients [65]. HSP70 can also inhibit apoptosis via Bcl-2, Bcl-X, Mcl-1, and others [67].

Regarding EMT, Cultrara et al. showed that the upregulation of glucose-regulated protein 78 (GRP78), which is a homolog of HSP70, reduces the E-cadherin expression in prostate cancer cells via the TGF-β1 pathway and, therefore, enhances the cell’s ability to pass through EMT [68]. Teng et al. showed that by decreasing HSP70 and HSP90, the tumor loses its ability to invade [69]. Hance et al. supported the findings by showing that prostate cancer cells treated with HSP90 promoted proliferation and expressed mesenchymal features [70]. In addition, Chatterjee et al. describe that HSP70 promotes the expression of Hif-1α and NF-κB as part of the regulation of metastasis [71].

Another interesting approach is that extracellular HSP90 (eHSP90) activates the MEK/ERK pathway that upregulates EZH2, which is the known EMT-related molecule involved in the molecular pathogenesis of CRPC [72,73]. Also, the authors found that eHSP90 inhibition resulted in cells that showed the phenotype of epithelial cells [74].

## 6. Extracellular Matrix Proteins and EMT in Prostate Cancer

### 6.1. Collagen Triple Helix Containing 1 Protein (CTHRC1)

Collagen triple helix repeat-containing 1 (CTHRC1) is a protein that was found in 2005 in the injured arteries of rats. The protein is associated with many physiological and pathological events, such as embryogenesis, liver cirrhosis, and rheumatoid arthritis [75]. CTHRC1 has a significant relation to EMT in prostate cancer [76]. 

Ma et al. showed on tissue microarray and further immunohistochemistry that prostate cancer cells express high levels of CTHRC1 and that blocking CTHRC1 demonstrated inhibition of invasiveness and aggressive characteristics of the tumor and upregulation of the epithelial morphological characteristics over the mesenchymal ones [76]. The authors also showed that CTHRC1 is tightly regulated by miRNA-30e-5p. There is a correlation between the expression of the gene and an increased proliferation, invasion, higher TNM staging, and a poor prognosis. Therefore, the authors suggest using CTHRC1 as a potential therapeutic target for prostate cancer [76].

Although the molecule is considered a possible target, not enough clinical trials were conducted to support this hypothesis. 

### 6.2. Periostin

Periostin is a TGF-β induced protein discovered in 1993. It is part of the extracellular matrix, secreted mainly into the extracellular environment. It can also be found in breast and prostate cancer epithelium, as was described in other works of our group [26]. It is normally expressed by healthy cells in small amounts while overexpressed in pathological states [77]. Also, periostin has a vital role in embryogenesis, maintenance of ECM, tissue repair, and others [76].

Tian et al. describe the connection between periostin and EMT initiation in prostate cancer. Periostin acts via PI3-Akt and other signaling pathways to promote EMT, angiogenesis, invasion, and metastasis [78].

Hu et al. described the possibility of initiating the EMT process through TGF-β-induced upregulation of periostin, STAT3, and Twist1. Inhibition of periostin in prostate cancer cells caused an inhibition of cellular proliferation. The researchers proved that high periostin expression is associated with the upregulation of N-cadherin and fibronectin and the downregulation of E-cadherin using Western blot analysis [79]. Therefore, the possible usage of Akt inhibitors is suggested and will be further discussed in Section 10.

## 7. Immune-Mediated EMT

The immune system plays a role in EMT; on the contrary, the EMT process can modulate the immune system. Liao et al. explain that the exact mechanism by which tumors can modify the immune system and EMT was not explored on a large scale. In his work, Liao describes a lymph node metastasis-associated suppressor (LNMAS) that suppresses metastasis in cervical cancer patients in vivo and in vitro. It was found that patients who were diagnosed with metastatic disease showed downregulation in LNMAS. Interestingly, LNMAS interacts with HMGB1, blocking the accessibility to TWIST1 and STC1, thus inhibiting the TWIST1 pathway and inhibiting EMT. Liao et al. used RNA-seq data to determine the exact dysregulated genes. The EMT-associated gene expression was validated using qPCR and Western blot. Another important aspect of this work showed the role of LNMAS downregulation in the immune escape of tumor cells from macrophage phagocytosis by downregulating STC1- an immune checkpoint molecule that usually precipitates in phagocytes. This phenomenon was verified by phagocytosis assays in vitro [80].

Another study is related to PCSK9, which promotes metastasis in colon cancer via EMT regulation. PCSK9 is a protein that usually binds to low-density lipoprotein receptors to regulate lipoprotein metabolism. PCSK9 inhibitors such as evolocumab and alirocumab are widely used in clinical practice as one of the treatment lines for preventing hypercholesterolemia-associated pathologies. Wang et al. showed that PCSK9 regulates EMT and PI3K/AKT signaling and phenotypic polarization of macrophages. He indicated that the knockdown of PCSK9 expression reduced colon cancer proliferation and metastasis by upregulation of Snail1 and, by this, downregulated E-cadherin expression while overexpressing N-cadherin. Proved by immunohistochemistry, Wang was able to show that low expression of PCSK9 is associated with low-grade tumors, while high expression is in high-grade tumors [81].

Regarding prostate cancer, a study by Fang et al. showed that a genetic inhibition of PCSK9 is associated with a lower risk of onset of prostate cancer. Two sample Mendelian randomization analyses were used to investigate the association of genetically proxied targets (PCSK9, HMGCR, and NPC1L1) with the risk of onset of prostate cancer [82]. On the other hand, a study by Gan et al. sought the relationship between PCSK9 and overall outcome in prostate cancer patients who were exposed to ionizing radiation. It has been shown that PCSK9 siRNA treatment protects prostate cancer cells exposed to radiation by reducing apoptosis and MMP inhibition [83]. While Sun et al. showed, as well, that PCSK9 inhibitors are strongly associated with a lower risk of prostate cancer [84], another cohort study did not find any correlation between PCSK9 expression and a high Gleason grade or a progression to lethal cancer [85]. 

In breast cancer, Datar and Schalper showed that increased vimentin/reduced E-cadherin is associated with PD-L1 upregulation and escape of the tumor from the immune system, while downregulation of PD-L1 correlates with low expression of mesenchymal phenotype in breast cancer. By this, they showed the relationship between EMT and PD-L1. In melanoma, Snail1-induced EMT is associated with impaired dendritic cell function and increased CD4+/FOXP3+ regulatory T cells. Also, the overexpression of β-catenin is related to the reduction of T-cell infiltration [86].

In prostate cancer, tumor-associated macrophages secrete pro-inflammatory cytokines such as CCL2 and CCL5, thus enhancing the migratory ability of prostate cancer cells to other parts of the body, especially the bone. In a study by Messex and Liou, CCL5 was added to a culture of prostate cancer cells, and thus, increased migratory and invasive abilities were observed. Moreover, high expression of CCL5 was correlated with a higher Gleason grade. Interestingly, CCL5 was found to activate STAT3, which is associated with upregulation of CCL2 and induction of EMT. Therefore, it was suggested that STAT3 plays a role in regulating oncogenic pathways and invasiveness [87].

## 8. Matrix Topology, Stiffness, Growth-Induced Solid Stress, and Low pH Enhance EMT

Extracellular matrix architectural topology and stiffness can activate cell migratory programs via different signaling pathways [26]. Matrix topology and stiffness are highly dependent on the mechanical environment, and stiff ECM induces elements of EMT through mechanotransduction [88,89]. The extracellular matrix structure is primarily influenced by collagen fibers that represent a major component of the ECM [90]. Fibroblasts, cancer-associated fibroblasts, and other cells produce collagens. Collagen crosslinking is an essential part of ECM stiffening. The main enzyme responsible for crosslinking is lysyl oxidase (LOX). Its inhibition caused decreased stiffness and increased tumor latency in breast, colorectal, lung, ovarian, and esophageal cancer tissues [91,92]. Kallikrein-related peptidase 4 (KLK4)-induced LOX overexpression and matrix remodelation in prostate cancer were shown by Kryza et al. [93].

CAFs are significant modifiers of the tumor microenvironment [94]. A review by Taddei et al. describes many significant adaptation changes involving CAFs. Interestingly, CAFs are undergoing Warburg metabolism, which allows prostate cancer survival. Warburg metabolism is an aerobic glycolytic pathway crucial for the tumor and initiates upon decreased environmental pH. The deregulated ‘reversed’ pH is an essential factor in the achievement of evasion of apoptosis, metabolic adaptation, migration, and invasion. Maintaining intracellular pH is vital for preventing damage to the cytosol and is regulated by HIF-1, which regulates ion pumps and transporters [95]. Giannoni et al. showed another interesting aspect of CAF, which is associated with enhanced expression of stem cell markers by promoting EMT and self-renewal and increasing the invasiveness, aggressiveness, and possible metastatic spread [96]. Another interesting factor in the microenvironment that has a significant impact on the tumor is the endothelial progenitor cells. Endothelial progenitor cells, originating from the peripheral blood and the bone marrow, contribute to tissue repair in physiological states and neoangiogenesis in response to VEGF secretion [95]. 

Growth-induced solid stress could be a further crucial environmental contributor to epithelial-mesenchymal transition. It was suggested that a growing tumor affects its environment while growing in two ways. The first is the mechanical-induced stress that the tumor exerts upon the surrounding tissues, and the second is growth-induced stress, which occurs when the cells strain structures [97]. Previously, it was suggested by Tse et al. that mechanical stress is the reason for cancer cell metastasis [98]. Chen et al. showed a different approach where mechanical stress did not induce EMT, although it enhanced cell migration. Combining IL-6 with compression triggered EMT in clear cell renal cell carcinoma (ccRCC). The EMT trigger was achieved due to the promotion of nuclear transcription of β-catenin by IL-6. The signaling pathway suggested as the one related to this phenomenon is the PI3K-Akt pathway [97]. 

## 9. EMT Studies Methods

Many methods are being used to understand better the EMT process and especially the underlying pathways that cause it. Several methods are described elsewhere in this article. In this chapter, we will describe the primary techniques used nowadays. Flow cytometry enables us to detect cell populations by the surface markers expressed on their membrane. In EMT, the markers are CD44, CD24, and ALDH. More precise utilization of flow cytometry over the years has led to the discovery of novel markers such as CD104, EpCAM, CD51, CD61, and CD106. The disadvantage of this technique is that it cannot detect the expression of some essential intracellular markers, such as vimentin and ZEB1 [99]. Another disadvantage is the limit of the technique; only live tissues can be used, and these may be unavailable in some hospitals. The advantage is that the process is relatively easy and fast [99]. Another technique that is being widely used is immunohistochemistry/immunofluorescence (IHC/IF). While the hematoxylin-eosin stain is used in histopathology for basic microscopical evaluation of potential pathologies, it does not allow us to obtain a complete record of receptors, which can be crucial for further treatment and prognosis. Tissue-based techniques such as IHC/IF give us information not only about the presence but also about their current location as well as possible colocalization within the cell [99], as shown by our group for the first time when we observed the colocalization of the EMT-regulator molecule slug and skp2 protein in high-grade prostate cancer epithelial cell cytoplasm [100]. Efforts to implement various markers in association with morphological changes for the prediction of EMT in vivo were unsuccessful, while in vitro, there was partial success [101].

On the contrary, the immunofluorescence microscopy assay of cytoskeletal remodeling elements successfully determines the transcriptional nodes that control EMT and may be a useful therapy target. Although combining immunofluorescence with other methods has an advantage, the limiting number of probes does not allow for describing the whole complexity of cells between mesenchymal and epithelial stages [102]. 

Transcript-based methods are used to understand the effect of genetic changes and EMT induction. One of them is RNA sequencing, which enables us to generate the exact EMT genetic sequence, which helps us to understand the precise entrance mechanism into EMT and connect it to the patient’s prognosis. The ability to induce EMT using this method and the knockdown of genes allowed the identification of signal cascades such as TEAD2, FOSL2, SP1, and other pathways not previously associated with EMT. While the advantages of the methods are understandable, the cost, processing duration, and inability to separate the tumor from the stroma make the approach impractical for everyday clinical practice. 

According to Brown et al., multiplexed image-based methods have many advantages. They are easy to perform, can be conducted with high throughput, and create a spatial model of the tumor, which will then be analyzed and processed. Brown’s laboratory used a multiplexed, multi-round tyramide signal amplification staining method using six canonical EMT markers to determine EMT score in a model of EMT in breast cancer patient samples. The technique successfully segmented out a tissue that plays a role in EMT, e.g., a fibroblast that surrounds tumors and frequently expresses vimentin, ZEB1, and other markers. The ability to do so dramatically impacts our understanding of how the tumor acts and its heterogenicity within EMT, prediction of the patient’s prognosis, and indication of the proper treatment [99]. 

## 10. EMT as a Therapeutic Target in Prostate Cancer

Targeting EMT is considered promising. It will restrict the tumor to its primary location, possibly prevent the need for systemic therapy required in metastatic disease, and make a more accessible and more successful resection. As shown earlier, targeted molecules can be transcription factors involved in EMT, such as Snail, Slug, and Twist, or EMT-related proteins, such as vimentin, N-cadherin, and others that may prevent tumor cells from metastasizing [103]. Qiao and Tian described that ATL-1 inhibits EMT by targeting Hsp27 and enhances the effect of cabozantinib in prostate cancer. Also, ATL-1 inhibited the further proliferation of prostate cancer and induced apoptosis [104]. 

Baritaki et al. described in their work the use of proteasome inhibitors in the suppression of Snail and reduction of RKIP in prostate cancer. They demonstrated that a stable Snail overexpression, Snail- 6SA, in a non-metastatic prostate cancer line results in EMT features acquisition. They also showed that DETANONOate (a nitric oxide donor) treatment reduces NF-κB DNA binding activity and Snail pathway inhibition, which mediates anti-metastatic properties in prostate cancer patients [105]. 

Another study by Fischer et al. showed that the inactivation of Snail and Twist had no effect on breast and pancreatic cancer progression in mouse models but affected the chemoresistance. They found evidence that the chemoresistance and metastatic potential in the same pancreatic cancer were conducted by ZEB1 and not by the molecules targeted originally. Thus, it is essential to point out that targeting single EMT driver molecules may be insufficient and may require targeting their combinations simultaneously and in relation to the type of tumor [106]. 

Castellón et al. wrote that SNAI/Slug upregulates the stemness of gene Sox2 and, by this, promotes metastasis and castration resistance in prostate cancer. Therefore, they proposed to study the impact of SNAI2/Slug axis manipulation on the outcomes of metastasis and resistance [107]. 

In BPH, Slabakova et al. described the upregulation of SNAI2/Slug and ZEB1 and changes in the expression of ZEB2 and miR-200. They observed that Slug is important for EMT initiation, while ZEB and miR-200 are crucial in resisting the reversal of TGF-β1- induced EMT. ZEB2 is responsible for downregulating the microRNAs of miR-200 and miR-205 and thus prevents the reversal of mesenchymal phenotype back to epithelial [108].

EMT plays a role in the progression from a tumor that responds to treatment into a castration-resistant one. As mentioned before, hypoxia affects the initiation of EMT. According to Gogola et al., hypoxia-inducible factor (HIF), particularly HIF-1, is associated with TGF-β signaling, among other pathways. This phenomenon was observed in prostate cancer and other tumors as well [109]. They also found out that there is a high expression of ZEB in prostate cancer, which is associated with a higher tumor grade, metastasis, and resistance to treatment. As part of the study, they checked various signaling pathways related to EMT, including the TGF-β, RTK, Wnt, Notch, and Hedgehog pathways (Figure 2), and these, in turn, were highly correlated with the promotion of prostate cancer growth and stemness properties. Although the analysis shows that epithelial plasticity correlates with poor clinical prognosis, the practical clinical use of EMT-related biomarkers has not yet been established [109]. 

There are many clinical trials in different stages for targeting EMT [109]—clinical trial no. NCT04021394 tries to capture circulating tumor cells (CTCs) in the blood of prostate cancer patients. According to previous trials, prostate cancers that passed EMT tend to shed cells higher than those possessing only the epithelial phenotype. The cellSearch^®^- (Menarini Silicon Biosystems Inc., Huntington Valley, PA, 19006, USA) based assay was used and captured CTCs only in an early stage of the disease in vivo. Still, a significant number of circulating cells could be collected only in the metastatic phase of the disease. The trial aimed to try and overcome the limitations of the current clinical assays and to obtain the full potential of the CTCs [110]. 

Another prospective clinical trial (no. NCT03381326) looked for biomarkers, such as CTCs, free DNA, stem cells, and EMT-related antigens, that may predict the outcome of cabazitaxel treatment in castration-resistant prostate cancer. All patients received cabazitaxel, and then a blood sample was taken for the presence of CTCs. In the case of disease progression, the researchers took more samples to detect the CTCs and tried to establish the correlation between CTCs and other markers and the stage of disease [111]. In an article following the current clinical trial, Koinis et al. described that after one treatment cycle with cabazitaxel, 75% of the patients had more than one detectable CTC/7.5 mL of peripheral blood and 60% with more than 5 CTCs/7.5 mL of peripheral blood. The detection of the CTCs correlated with poor overall survival rates. Therefore, they suggest that in patients with mCRPC treated with cabazitaxel, the CTC count at baseline and after the first treatment may be used to establish further prognosis and efficacy [112].

Ramesh et al. covered in an article a few drugs that may be potentially used for inhibiting the EMT process in different types of cancer. Simvastatin, a common drug used for hyperlipidemia by inhibiting HMG CoA reductase, has been found to interfere with the EMT process in advanced colorectal cancer, pancreatic cancer, small cell lung cancer, and brain metastasis. The drug was checked in a few clinical trials. It was found to downregulate various mesenchymal markers, such as vimentin, while upregulating epithelial markers, such as E-cadherin, in carcinoma of the urinary bladder. Also, the downregulation of β-catenin was inspected in some lines of bladder cancer upon treatment with simvastatin. In pancreatic adenocarcinoma, pre-surgical treatment with simvastatin was associated with lower expression of CXCR4, c-Met, and vimentin [113]. On the contrary, in vivo prostate cancer cell lines showed an increase in aggressiveness as a result of treatment with simvastatin [114]. Other drugs targeting EMT in different oncological diagnoses are olaparib, used mainly in BRCA-positive breast cancer patients, disulfiram, etodolac, suramin, and others [113].

A clinical trial first submitted In 1999 checked the possible targeting of EMT by suramin in prostate cancer, NSCLC, and other advanced solid tumors. Suramin is a well-known drug indicated for the treatment of African sleeping sickness. It inhibits the EMT process by inhibiting heparanase, an enzyme that modulates the tumor’s stroma to express TGF-β and promote EMT. The mentioned trial no. NTC00002881 checked the combination of suramin with flutamide and hydrocortisone after orchiectomy or LHRH analog treatment versus patients treated with flutamide after orchiectomy or LHRH analog in patients with histologically confirmed prostate adenocarcinoma with bone metastasis but without brain metastasis and/or spinal cord compression. In an article published following the trial by Small et al., suramin was shown to be efficient in the palliative setting by slowing the progression of the disease but did not affect the overall survival rate [115].

In another article by Gracia-Schurmann et al., treatment with suramin decreased PSA over time without significant toxicity in 33% of the patients, while 48% did not show a change in PSA. The suramin-responsive group showed an average survival of 495 days, the stable group demonstrated survival of 341 days, and the nonresponsive group showed the lowest survival of 79 days [116]. 

Regarding docetaxel-resistant prostate cancer, many treatment possibilities were suggested. Ganju et al. describe using nanoparticles to overcome the drug’s increased efflux and low influx. Moreover, the nanoformulation decreases the needed dose of chemotherapy due to more precise targeting. By this, the adverse reactions profile of the chemotherapeutics was relatively low. Also, it is possible to combine other targeted therapies within the nanoparticle to improve the outcome [117]. Nanoparticles containing docetaxel conjugated to miRNA2c, which is a direct target of ERG and a critical inhibitor of EMT and prostate cancer cell motility, have increased bioavailability and inhibited not only cancer cells but also cancer stem cells [117,118].

HSPs are also a promising target for EMT inhibition in prostate cancer. According to Chatterjee et al., phenylethynesulfonamide (PES), also known as pifithrin-µ, is a small molecule that inhibits HSP70, leading to intracellular instability and promoting apoptosis. Nevertheless, the molecule is not an FDA-approved drug [71]. 

A clinical trial no. NCT01685268 is an open-label, randomized trial conducted in 2016 by Slovin et al. that checked the possible utilization of onalespib, an HSP90 inhibitor. They administered the drug alone or in combination with abiraterone acetate in patients with CRPC who were no longer responding to abiraterone alone. The results showed no clinical improvement, although some mild biological effect was gained. Therefore, they decided not to continue further research on this drug [119]. 

HSP90—an essential member of the chaperone family—promotes EMT and metastatic progression through different mechanisms involved in cell signaling, growth, and survival [120]. According to Chen et al., ganetespib, an HSP90 inhibitor, is a promising treatment line for patients with prostate cancer. It has been shown that the combination of ganetespib with enzalutamide (an AR inhibitor) caused cell death. Also, the combination led to a reduction in AR activity. They state that further research is required for this treatment [121]. A phase II clinical trial no. NCT01270880 was reported by Thakur et al. and showed a minimal clinical impact of ganetespib on patients with mCRPC [122].

Another interesting drug that may affect the EMT process in prostate cancer is metformin. Metformin is a biguanide, considered until recent years as the first line treatment of diabetes mellitus type 2. The exact mechanism of action of metformin is obscure [123]. According to Wang et al., metformin suppresses EMT via inactivation of the NF-κB signaling pathway. They showed that suppression of TNF-α by metformin decreased the ability of the tumor to invade and cause metastasis [124]. Tong et al. showed that metformin can repress EMT via downregulating COX2, PGE2, and STAT3 levels [125]. Therefore, metformin may be combined with other chemotherapeutics as a potential therapy for CRPC—clinical trial no. NCT04033107 tries to combine metformin with ascorbate (vitamin C) as a treatment possibility for many kinds of cancers. In BPH, Li et al. examined the relationship between ascorbate and BPH due to the known property of ascorbate to inhibit HIF-α. Notably, it was revealed by the researchers that the HIF-α is destabilized by vitamin C via hydroxylation of prolyl. Vitamin C decreased VEGF and prostate HIF-α levels in rats and was suggested for further investigation as a promising treatment line for BPH [126]. 

Over the years, the Akt pathway has received significant attention, and much effort has been put into creating Akt inhibitors for downregulating periostin. In recent years, Phase II and III clinical trials have been conducted on Akt inhibitors such as ipatasertib and capivasertib. The trials showed positive results in metastatic CRPC and PTEN loss. The drugs inhibit the PI3K-Akt-mTOR pathway, which, as presented earlier, is essential in developing prostate cancer cells. The Phase II clinical trial no. NCT01485861 compared treatment with ipatasertib vs. therapy with a placebo in patients treated with abiraterone and who were previously treated with docetaxel. The group treated with ipatasertib showed improved radiological progressive-free survival [127]. An ongoing clinical trial no. NCT04493853 currently checks the treatment with a combination of capivasertib, abiraterone, prednisone, and androgen deprivation therapy (ADT) vs. a combination of abiraterone, prednisone, and ADT. The trial showed a decrease in PSA levels in one-third of the patients. Nevertheless, further studies are still required [127,128].

As described earlier, silibinin can inhibit the TGF-β signaling pathway and thus downregulate CAF and EMT. Clinical trial NCT00487721 tried to prove this hypothesis. Although clinically, PSA levels were not found to decrease over time in the patients, a preclinical evaluation of cancer cells showed a positive result when chemotherapeutic treatment was combined with silibinin [129]. Several drugs targeting EMT with or without FDA approval are listed in Table 1.

## 11. Conclusions

In conclusion, EMT is a fundamental process that occurs in pathological states and physiological processes such as organogenesis during embryonic life and wound healing. In many tumors, and especially in prostate cancer, it seems that EMT plays a crucial role in the creation of a supporting microenvironment for the tumor to thrive and eventually metastasize. The transition process from static epithelium to mesenchyme is one of the key factors in classifying a tumor as a high-grade tumor that can send metastasis. 

Further understanding for early detection of possible metastatic tumors or its restriction to the primary locus may turn the tumor into an easily resectable, less radio-resistant, and chemo-resistant. The complex process involves many genes and transcription factors such as Twist, Snail, Slug, Zeb, and more. Although some clinical trials check the possible application of the knowledge about the EMT process, more research is still required. Nevertheless, EMT pathways remain promising potential targets for therapy, not merely in prostate cancer but in many other types of cancer as well. 

## Figures and Tables

**Figure 1 biomedicines-12-00418-f001:**
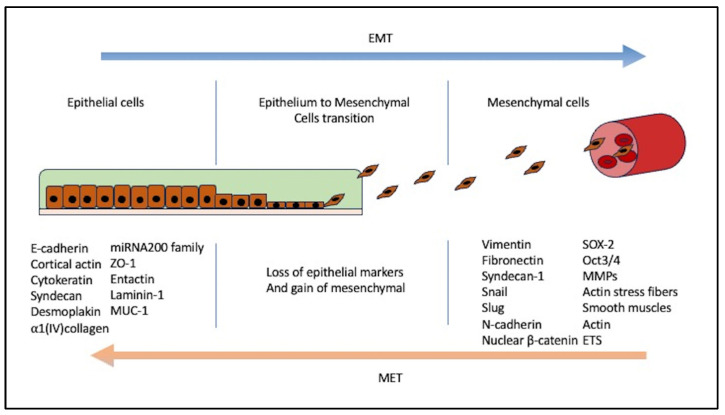
Epithelial mesenchymal transition. The process involves a transition of extracellular matrix components, anchoring the cell to the basement membrane. Losing epithelial anchors co-occurs with the gain of mesenchymal markers, indicating cells that have passed through EMT. Proteins that are downregulated during EMT are E-cadherin, β-catenin, laminin, desmoplakin, Muc-1, ZO-1, syndecan-1, cytokeratin 18, and the newly studied EPLIN [2]. Proteins gained are, for example, Snail, Twist, Slug, LEF-1, Scratch, SIP1, E47, Ets, FTS binding protein, RhoB, FSP1 (S100A4), TGF-beta, FGF-1,-2,-8, MMP-2, MMP-9, Vimentin, αSMA, FOXC2, fibronectin, collagen type 1, collagen type III, thrombospondin, PAI-1, etc. [2].

**Figure 2 biomedicines-12-00418-f002:**
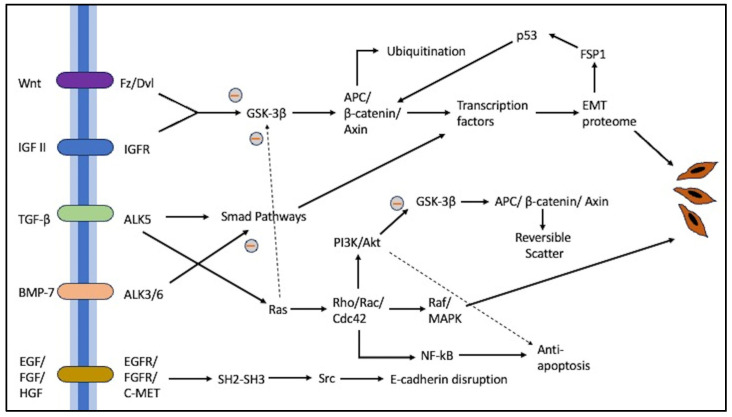
Pathological EMT pathway and its association with E-cadherin repression. Epithelial plasticity can lead to EMT induction, as described earlier. These events are regulated tightly by intrinsic kinase receptors on the cell surface, which intracellularly are further modulated by various molecules such as GTPases, Smads, PI3Ks, and MAP kinases, and thus alter the amount of β-catenin available for further EMT induction processes in the nucleus. Levels of β-catenin are regulated by E-cadherin or APC/β-catenin/Axin complexes, which transfer β-catenin between ubiquitination or utilization in adherens junctions. The activation of transcription causes the production of other factors, such as Snail SIP1, etc., which are essential for EMT regulation. The concentration and types of cell surface receptors and kinases determine the individual characteristics of each tissue type. Other important EMT pathways work for cytoskeleton rearrangement (e.g., RTK-JAK-ERK axis) or focal adhesion and stress fiber formation, such as the RhoA-vinculin axis (not shown here).

**Table 1 biomedicines-12-00418-t001:** List of drugs targeting EMT. The drugs are FDA-approved or under a clinical trial.

Drug	Target	Clinical Use	FDA-Approval	Trial	Clinical Trial ID
Suramin	Heparanase	Trypanosomiasis	+	Prostate cancer, advanced solid tumors	NCT00002881
DETANONOate	NF-κB, Snail	Not currently used. Under trials in many fields	−	Prostate cancer	-
Cabazitaxel	MMP-9	Prostate cancer	+	Novel biomarkers in cabazitaxel treated patients with prostate cancer	NCT03381326
Onalespib	HSP-90	Not currently used	−	Prostate cancer	NCT01685268
Ganetespib	HSP-90	Not currently used	−	Prostate cancer	NCT01270880
Metformin	NF-κB, PGE2 STAT3, COX2	Type 2 DM	+	Combination with ascorbate for BPH	NCT04033107
Ipatasertib	PI3K-Akt-mTOR	Not currently used	−	Prostate cancer	NCT01485861
Capivasertib	PI3K-Akt-mTOR	HR+, HER2- breast cancer	+	Prostate cancer, Triple-negative breast cancer	NCT04493853
Ascorbate	HIF-α	Supplement	+	Combination with metformin for BPH	NCT04033107
Silibinin	TGF-β	Toxic liver damage	+	Combination with chemotherapies for prostate cancer	NCT00487721

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
