# Peer review of "Epithelial-Mesenchymal Transition: A Fundamental Cellular and Microenvironmental Process in Benign and Malignant Prostate Pathologies"

_biomedicines, 2024, doi:10.3390/biomedicines12020418_

Round 1

Reviewer 1 Report

Comments and Suggestions for Authors

The manuscript is a review article on the role of epithelial-to-mesenchymal transition (EMT) in the microenvironment process in benign and malignant prostate pathologies. The authors have attempted to organize the review by describing various types of EMT and their role in benign and malignant prostate, characterization of cellular and tissue morphology, mechanism(s) of EMT induction and molecular switch highlighting the role of some key members of the process. The authors also touch base on the immune-mediated EMT effects, tissue changes and finally discussion on the therapeutic application. The review touch base on several aspects, however none of them are described in detail. For example, the description of various types of EMT could be better represented with figures on each type. The cellular and tissue morphology lacks images of the EMT in the benign and malignant prostate. The EMT process is not covered in greater details and lack description about the signaling pathways involved in the EMT process. Moreover, the therapeutic implication does not describe the use of various drugs that can combat the EMT process and also the clinical trials on various markers and cell types having ability to prognosticate EMT. Overall, the review article is premature and requires extensive write-up on each sub-section.

Author Response

Dear Reviewer,
On behalf of the co-authors, I would like to thank you for the time you dedicated to reading our review.
Please find our responses to the comments.
We agree that some sections' depth is basic; therefore, when it was relevant, we added much more information. For example, we added more EMT induction mechanisms and a few new sections about Heat shock proteins (HSPs) and extracellular matrix proteins related to EMT, such as Periostin and CTHRC1,
which are related to prostate cancer. In some cases, it was impossible to refer to prostate pathologies at all because, e.g., Type 1 and Type 2 EMT are different processes that have nothing to do with prostate cancer.
Still, they are crucial for the complete understanding of the topic. Moreover, we think that it is not practical to expand the information about each protein that is related to EMT. Nevertheless, we tried to rewrite the review to be more clinically relevant.
Also, you mentioned the need for more figures in the sections that describe the different types of EMTs. We think that adding more figures will be overwhelming and will not serve its role in explaining the EMT process in prostate cancer. The figure that adequately describes Type 3 EMT is Figure 1.
Regarding the therapeutic implications- we agree with the statement; therefore, we drastically expanded the possible targets and provided a few drugs approved by the FDA or under clinical trial and their targets. All the drugs and trials are described in the text and a table, which will be a comfortable summarization tool.
About the tissue morphology images- The type and extent of the review article do not allow us to include our previously published or unpublished experimental data where we document the association of tissue morphology with molecular profiles of cells in epithelial or mesenchymal state. Nevertheless, a schematic
illustration of the morphological changes is included in Figure 1.
Several times in the article, we mention the role of different signaling pathways in the EMT process. New information from several recent articles was also included to describe the role of key signaling pathways in the EMT process.
We have significantly modified the manuscript by adding more specific and detailed information regarding the EMT process in benign prostate disease and prostate cancer to each sub-section. We have also commented in a relevant way if there was insufficient experimental and Clinical evidence about a particular
topic.
We hope we have addressed each comment and answered all issues satisfactorily.
On behalf of the co-authors, I would like to thank you again for your precious time and hope for further cooperation.

Reviewer 2 Report

Comments and Suggestions for Authors

In this well-organized review, Goncharov et al focus the role of EMT in benign prostate disease and in the carcinogenesis process, supporting with good references,  the statement that transition process from static epithelium to mesenchyme is one of the key factors in classifying a tumor as a high-grade tumor that can send metastasis. I suggest to replace figure 1 and 2 with more complex ones and modifying their description.

Author Response

Dear Reviewer,
On behalf of the co-authors, I would like to thank you for the time you dedicated to reading our review.
Please find our responses to the comments.
As suggested, the figures' descriptions were modified, but we decided not to change the figures because they describe the fundamentals of the EMT process straightforwardly and clearly. We truly believe that changing or over-complicating the figures will not assist the role of this review in describing the process. Figure 1 contains all the proteins gained or lost during the process. Figure 2 explains the EMT pathways and their association with E-cadherin repression.
We hope we have addressed the issues satisfactorily.
On behalf of the co-authors, I would like to thank you again for your precious time and hope for further cooperation.

Round 2

Reviewer 1 Report

Comments and Suggestions for Authors

In the revised version the authors have elaborated the text and have included information on various types of EMT, highlighted the role of some key members of the EMT process and adding details on the therapeutic aspects of the EMT. The review touch base several key aspects of the EMT process in prostate cancer and benign disease. The revised manuscript is significantly improved from its previous version and is acceptable for publication.